# Robustness against Relational Adversary

## Abstract

Test-time adversarial attacks have posed serious challenges to the robustness of machine-learning models, and in many settings the adversarial perturbation need not be bounded by small $\ell_p$-norms. Motivated by the semantics-preserving attacks in vision and security domain, we investigate *relational adversaries*, a broad class of attackers who create adversarial examples that are in a reflexive-transitive closure of a logical relation. We analyze the conditions for robustness and propose *normalize-and-predict* – a learning framework with provable robustness guarantee. We compare our approach with adversarial training and derive an unified framework that provides benefits of both approaches. Guided by our theoretical findings, we apply our framework to image classification and malware detection. Results of both tasks show that attacks using relational adversaries frequently fool existing models, but our unified framework can significantly enhance their robustness.

## 1 Introduction

The robustness of machine learning (ML) systems has been challenged by test-time attacks using adversarial examples (Szegedy et al., 2013). These adversarial examples are intentionally manipulated inputs that preserve the essential characteristics of the original inputs, and thus are expected to have the same test outcome as the originals by human standard; yet they severely affect the performance of many ML models across different domains (Moosavi-Dezfooli et al., 2016; Eykholt et al., 2018; Qin et al., 2019). As models in high-stake domains such as system security are also undermined by attacks (Grosse et al., 2017; Rosenberg et al., 2018; Hu & Tan, 2018), robust ML in adversarial test environment becomes an imperative task for the ML community.

Existing work on test-time attacks predominately considers $\ell_p$-norm bounded adversarial manipulation (Goodfellow et al., 2014; Carlini & Wagner, 2017). However, in many security-critical settings, the adversarial examples need not respect the $\ell_p$-norm constraint as long as they preserve the malicious semantics. In malware detection, for example, a malware author can implement the same function using different APIs, or bind a malware within benign softwares like video games or office tools. The modified malware preserves the malicious functionality despite the drastically different syntactic features. Hence, focusing on adversarial examples of small $\ell_p$-norm in this setting will fail to address a sizable attack surface that attackers can exploit to evade detectors.

In addition to security threats, another rising concern on ML models is the spurious correlations they could have learned in a biased data set. Ribeiro et al. (2016) show that a highly accurate wolf-vs-husky-dog classifier indeed bases its prediction on the presence/absence of snow in the background. A reliable model, in contrast, should be robust to changes of this nature. Although dubbed as semantic perturbation or manipulation (Mohapatra et al., 2020; Bhattad et al., 2019), these changes do not alter the core of the semantics of input data, thus, we still consider them to be semantics-preserving pertaining to the classification task. Since such semantics-preserving changes often resulted in large $\ell_p$-norms, they are likely to render the existing $\ell_p$-norm based defenses ineffective.

In this paper, we consider a general attack framework in which attackers create adversarial examples by transforming the original inputs via a set of rules in a semantics-preserving manner. Unlike the prior works (Rosenberg et al., 2018; Hu & Tan, 2018; Hosseini et al., 2017; Hosseini & Poovendran, 2018) which investigate specific adversarial settings, our paper extends the scope of attacks to general logical transformation: we unify the threat models into a powerful relational adversary, which can readily incorporate more complex input transformations.

From the defense perspective, recent work has started to look beyond $\ell_p$-norm constraints, including adversarial training (Grosse et al., 2017; Rosenberg et al., 2019; Lei et al., 2019), verification-loss regularization (Huang et al., 2019) and invariance-induced regularization (Yang et al., 2019). Adversarial training in principle can achieve high robust accuracy when the adversarial example in the training loop maximizes the loss. However, finding such adversarial examples is in general NP-hard (Katz et al., 2017), and we show in Sec 4 that it is even PSPACE-hard for semantics-preserving attacks that are considered in this paper. Huang et al. (2019) and Yang et al. (2019) add regularizers that incorporate model robustness as part of the training objective. However, such regularization can not be strictly enforced in training, and neither can the model robustness. These limitations still cause vulnerability to semantics-preserving attacks.

***Normalize-and-Predict Learning Framework*** This paper attempts to overcome the limitations of prior work by introducing a learning framework that guarantees robustness by design. In particular, we target a *relational* adversary, whose admissible manipulation is specified by a logical relation. A logical relation is a set of input pairs, each of which consists of a source and target of an atomic, semantics-preserving transformation. We consider a strong adversary who can apply an arbitrary number of transformations. Our paper makes the following contribution towards the theoretical understanding of robust ML against relational adversaries:

1. We formally describe admissible adversarial manipulation using logical relations, and characterize the necessary and sufficient conditions for robustness to relational adversaries.

2. We propose *normalize-and-predict* (hereinafter abbreviated as *N&P*), a learning framework that first converts each data input to a well-defined and unique normal form and then trains and classifies over the normalized inputs. We show that our framework has guaranteed robustness, and characterize conditions to different levels of robustness-accuracy trade-off.

3. We compare *N&P* to the popular adversarial training framework, which directly optimizes for accuracy under attacks. We show that *N&P* has the advantage in terms of explicit robustness guarantee and reduced training complexity, and in certain cases yields the same model accuracy as adversarial training. Motivated by the comparison, we propose a unified framework, which selectively normalizes over relations that tend to preserve the model accuracy and adversarially trains over the rest. Our unified approach gets the benefits from both frameworks.

We then apply our theoretical findings to malware detection and image classification. For the former, first, we formulate two types of common program transformation — (1) addition of redundant libraries and API calls, and (2) substitution of equivalent API calls — as logical relations. Next, we instantiate our learning framework to these relations, and propose two generic relational adversarial attacks to determine the robustness of a model. Finally, we perform experiments over *Sleipnir*, a real-world WIN32 malware data set. Regarding image classification, we reused an attack method proposed by the prior work (Hosseini & Poovendran, 2018) — shifting of the hue in the HSV color space — that can be deemed as a specific instantiation of our attack framework. We then compare the accuracy and robustness of ResNet-32 (He et al., 2016), a common image classification model, trained with the unified framework against the standard adversarial training on CIFAR-10 (Krizhevsky et al., 2009). The results we obtained in both tasks show that:

1. Attacks using addition and substitution suffice to evade existing ML malware detectors.

2. Our unified approach using input normalization and adversarial training achieves highest robust accuracy among all baselines in malware detection. The drop in accuracy on clean inputs is small and the computation cost is lower than pure adversarial training.

3. When trained with the unified learning framework, ResNet-32 achieves similar clean accuracy but significantly higher robust accuracy than adversarial training alone.

Finally, based on our theoretical and empirical results, we conclude that input normalization is vital to robust learning against relational adversaries. We believe techniques that can improve the quality of normalization are promising directions for future work.

## 2 RELATED WORK.

Test-time attacks using adversarial examples have been extensively studied in the past several years. Research has shown ML models are vulnerable to such attack in a variety of application domains (Moosavi-Dezfooli et al., 2016; Chen et al., 2017; Papernot et al., 2017; Eykholt et al., 2018; Ebrahimi et al., 2018; Qin et al., 2019; Yang et al., 2020) including system security where reliable defense is absolutely essential. For instance, Grosse et al. (2017) and Al-Dujaili et al. (2018) evade API/library usage based malware detectors by adding redundant API calls; Rosenberg et al. (2018), Hu & Tan (2018), and Rosenberg et al. (2019) successfully attack running-time behavior based detectors by adding redundant execution traces; Pierazzi et al. (2020) extend the attacks from feature-space to problem-space, propose a framework to describe real-world attacker's constraints and create realistic attack instances using automated software transplantation.

On the defense end, the work closest to ours in spirit is Yang et al. (2019), which adds invariance-induced regularizers to the training process. Their work however differs from ours in two major ways. First, their work considers a specific spatial transformation attack in image classification; our work considers a general adversary based on logic relations. Second, their regularizer may not enforce the model robustness on finite samples as they are primarily interested in enhancing the model accuracy. In contrast, our framework emphasizes robustness which is enforced by design. Grosse et al. (2017); Al-Dujaili et al. (2018); Rosenberg et al. (2019) improve robustness via adversarial training; we show such approach is hard to optimize. Incer et al. (2018); Kouzemtchenko (2018) enforce monotonicity over model outputs so that the addition of feature values always increase the maliciousness score. These approaches are limited to guarding against the addition attacks, thus lacks generality. Last, Xu et al. (2017) use feature squeezing, which quantizes the feature values in order to reduce the number of adversarial choices. However, their defense is for $\ell_p$-norm adversaries and thus inapplicable for relational attacks.

Normalization is a technique to reduce the number of syntactically distinct instances. First introduced to network security in the early 2000s in the context of intrusion detection systems (Handley et al., 2001), it was later applied to malware detection (Christodorescu et al., 2007; Coogan et al., 2011; Bichsel et al., 2016; Salem & Banescu, 2016; Baumann et al., 2017). Our work addresses the open question whether normalization is useful for ML under relational adversary by investigating its impact on both model robustness and accuracy.

## 3 BACKGROUND

In this section, we first describe the learning task, then formalize the potential adversarial manipulation as logical relations, and eventually derive the notion of robustness to relational adversaries.

***Learning Task.*** We consider a data distribution $\mathcal{D}$ over a input space $\mathcal{X}$ and categorical label space $\mathcal{Y}$. We use bold face letters, e.g. $\mathbf{x}$, for input vectors and $y$ for the label. Given a hypothesis class $\mathcal{H}$, the learner wants to learn a classifier $f : \mathcal{X} \to \mathcal{Y}$ in $\mathcal{H}$ that minimizes the risk over the data distribution. In non-adversarial settings, the learner solves $\min_{f \in \mathcal{H}} \mathbb{E}_{(\mathbf{x},y) \sim \mathcal{D}} \ell(f, \mathbf{x}, y)$, where $\ell$ is a loss function. For classification, $\ell(f, \mathbf{x}, y) = \mathbb{1}(f(\mathbf{x}) \neq y)$.

***Logical Relation.*** A relation $\mathcal{R}$ is a set of input pairs, where each pair $(\mathbf{x}, \mathbf{z})$ specifies a transformation of input $\mathbf{x}$ to output $\mathbf{z}$. We write $\mathbf{x} \to_{\mathcal{R}} \mathbf{z}$ iff $(\mathbf{x}, \mathbf{z}) \in \mathcal{R}$. We write $\mathbf{x} \to_{\mathcal{R}}^* \mathbf{z}$ iff $\mathbf{x} = \mathbf{z}$ or there exists $\mathbf{z}_0, \mathbf{z}_1, \cdots, \mathbf{z}_k$ ($k > 0$) such that $\mathbf{x} = \mathbf{z}_0$, $\mathbf{z}_i \to_{\mathcal{R}} \mathbf{z}_{i+1}$ ($0 \leq i < k$) and $\mathbf{z}_k = \mathbf{z}$. In other words, $\to_{\mathcal{R}}^*$ is the reflexive-transitive closure of $\to_{\mathcal{R}}$. We describe an example relation as follows:

**Example 1** (Hue Shifting). *Let $\mathbf{x}_h$, $\mathbf{x}_s$, $\mathbf{x}_v$ denote the hue, saturation and value components of an image $\mathbf{x}$. In a hue shifting relation $\mathcal{R}$, $\mathbf{x} \to_{\mathcal{R}} \mathbf{z}$ iff $\mathbf{z}_h = (\mathbf{x}_h + \delta) \% 1$ where $\delta$ is a scalar, $\mathbf{z}_s = \mathbf{x}_s$, $\mathbf{z}_v = \mathbf{x}_v$. Since $\mathbf{x}_h$ changes in a circle, i.e., hue of 1 is equal to hue of 0. Hence, we compute the modulo of the hue component with 1 to map $\mathbf{z}_h$ within [0,1] (Appendix B gives the background of HSV).*

In this paper, we also consider unions of relations. Notice that a finite union $\mathcal{R}$ of $m$ relations $\mathcal{R}_1, \cdots, \mathcal{R}_m$ is also a relation, and $\mathbf{x} \to_{\mathcal{R}} \mathbf{z}$ *iff* $\mathbf{x} \to_{\mathcal{R}_i} \mathbf{z}$ for any $i \in \{1, \cdots, m\}$.

Table 1: Comparison of training objective and test output for standard risk minimization learning scheme, *N&P* and adversarial training; $f^*$ is the minimizer of the training objective.

| | No Defense | Normalize-and-Predict | Adversarial Training |
|---|---|---|---|
| Train | $\min\limits_{f} \sum\limits_{(\mathbf{x},y)\in D} \ell(f,\mathbf{x},y)$ | $\min\limits_{f} \sum\limits_{(\mathbf{x},y)\in D} \ell(f,\mathcal{N}(\mathbf{x}),y)$ | $\min\limits_{f} \max\limits_{A(\cdot)} \sum\limits_{(\mathbf{x},y)\in D} \ell(f,A(\mathbf{x}),y)$ |
| Test | $f^*(x)$ | $f^*(\mathcal{N}(x))$ | $f^*(x)$ |

***Threat Model.*** A test-time adversary replaces a clean test input $\mathbf{x}$ with an adversarially manipulated input $A(\mathbf{x})$, where $A(\cdot)$ represents the attack algorithm. We consider an adversary who wants to maximize the classification error rate: $\mathbb{E}_{(\mathbf{x},y)\sim\mathcal{D}}\, \mathbb{1}(f(A(\mathbf{x}))\neq y)$.

We assume *white-box* attacks[1], i.e. the adversary has total access to $f$, including its structures, model parameters and any defense mechanism in place. To maintain the malicious semantics, the adversarial input $A(\mathbf{x})$ should belong to a feasible set $\mathcal{T}(\mathbf{x})$. In this paper, we focus on $\mathcal{T}(\mathbf{x})$ that is described by relation. We consider a logical relation $\mathcal{R}$ that is known to both the learner and the adversary, and we define a relational adversary as the following.

**Definition 1** (relational adversary). *An adversary is said to be $\mathcal{R}$-relational if $\mathcal{T}(\mathbf{x}) = \{\mathbf{z}\,|\,\mathbf{x} \to_{\mathcal{R}}^* \mathbf{z}\}$, i.e. each element in $\mathcal{R}$ represents an admissible transformation, and the adversary can apply arbitrary number of transformation specified by $\mathcal{R}$.*

We can then define the robustness of a classifier $f$ by how often its prediction is consistent under attack, and robust accuracy as the fraction of predictions that are both robust and accurate.

**Definition 2** (Robustness and robust accuracy). *Let $Q(\mathcal{R}, f, \mathbf{x})$ be the following statement: $\forall \mathbf{z}((\mathbf{x} \to_{\mathcal{R}}^* \mathbf{z}) \Rightarrow f(\mathbf{x}) = f(\mathbf{z}))$. Then, a classifier $f$ is robust at $\mathbf{x}$ if $Q(\mathcal{R}, f, \mathbf{x})$ is true, and the robustness of $f$ to an $\mathcal{R}$-relational adversary is: $\mathbb{E}_{\mathbf{x}\sim\mathcal{D}_{\mathcal{X}}}\, \mathbb{1}_{Q(\mathcal{R},f,\mathbf{x})}$, where $\mathbb{1}_{(\cdot)}$ indicates the truth value of a statement and $\mathcal{D}_{\mathcal{X}}$ is the marginal distribution over inputs. The robust accuracy of $f$ w.r.t. an $\mathcal{R}$-relational adversary is then: $\mathbb{E}_{(\mathbf{x},y)\sim\mathcal{D}}\, \mathbb{1}_{Q(\mathcal{R},f,\mathbf{x})\wedge f(\mathbf{x})=y}$.*

Notice that the robust accuracy of a classifier is no more than the robustness in value because of the extra requirement of $f(\mathbf{x}) = y$. Meanwhile, a classifier with the highest robustness accuracy may not always have the highest robustness and vice versa: an intuitive example is that a constant classifier is always robust but not necessarily robustly accurate. In Sec 4, we will discuss both objectives and characterize the trade-off between them.

## 4 *N&P* – A PROVABLY ROBUST LEARNING FRAMEWORK

In this section, we introduce *N&P*, a learning framework which learns and predicts over normalized training and test inputs. We first identify the necessary and sufficient condition for robustness, and propose a normalization procedure that makes *N&P* provably robust to $\mathcal{R}$-relational adversaries. Finally, we analyze the performance of *N&P*: since *N&P* guarantees robustness, the analysis will focus on robustness-accuracy trade-off and provide an in-depth understanding to causes of such trade-off.

### 4.1 AN OVERVIEW OF THE *N&P* FRAMEWORK

In *N&P*, the learner first specifies a normalizer $\mathcal{N} : \mathcal{X} \to \mathcal{X}$. We call $\mathcal{N}(\mathbf{x})$ the 'normal form' of input $\mathbf{x}$. The learner then both trains the classifier and predicts the test label over the normal forms instead of the original inputs. Let $D$ denote the training set. In the empirical risk minimization learning scheme, for example, the learner will now solve the following problem

$$\min_{f\in\mathcal{H}} \sum_{(\mathbf{x},y)\in D} \ell(f,\mathcal{N}(\mathbf{x}),y), \tag{1}$$

and use the minimizer $f^*$ as the classifier. During test-time, the model will predict $f^*(\mathcal{N}(\mathbf{x}))$. Table 1 compares the *N&P* learning pipeline to normal risk minimization and adversarial training.

---

[1]We consider a strong white-box attacker to avoid interference from security by obscurity, which is shown fragile in various other adversarial settings (Carlini & Wagner, 2017).

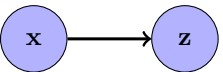 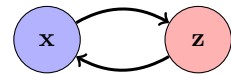 

Figure. 1: Relations with different robustness-accuracy trade-off. Different node colors indicate different most likely labels. Appendix A.7 gives a detailed explanation on why semantics-preserving transformation can still change the labels of data. **Left:** *N&P* preserves natural accuracy; **Middle:** *N&P* preserves robust accuracy; **Right:** *N&P* causes suboptimal robust accuracy: suppose $\mu(\mathbf{x}) = 0.02$, $\mu(\mathbf{z}_1) = \mu(\mathbf{z}_2) = 0.49$, and $\eta$ is deterministic. *N&P* predict the same label and thus has accuracy at most $0.49$, while the highest robust accuracy is $0.98$ by predicting the true label for $\mathbf{z}_1$ and $\mathbf{z}_2$.

## 4.2 FINDING THE NORMALIZER

The normalizer $\mathcal{N}$ is crucial for achieving robustness: intuitively, if $\mathbf{x}$ and its adversarial example $\mathbf{x}_{adv}$ share the same normal form, then the prediction will be robust. Meanwhile, a constant $\mathcal{N}$ is robust, but has no utility as $f(\mathcal{N}(\cdot))$ is also constant. Therefore, we seek an $\mathcal{N}$ that perform only the necessary normalization for robustness and has minimal impact on accuracy.

We first construct the **relational graph** $G_{\mathcal{R}} = \{V, E\}$ of $\mathcal{R}$: the vertex set $V$ contains all elements in $\mathcal{X}$; the edge set $E$ contains an edge $(\mathbf{x}, \mathbf{z})$ *iff* $(\mathbf{x}, \mathbf{z}) \in \mathcal{R}$. Then, a directed path exists from $\mathbf{x}$ to $\mathbf{z}$ *iff* $\mathbf{x} \rightarrow_{\mathcal{R}}^* \mathbf{z}$. We derive the following necessary and sufficient condition for robustness under *N&P* in Observation 1, and thus obtain a normalizer $\mathcal{N}$ in Proposition 1 that guarantees robustness.

**Observation 1** (Condition for Robustness)**.** *Let $C_1, \cdots, C_k$ denote the weakly connected components (WCC) in $G_{\mathcal{R}}$. A classifier $f$ is robust for all $\mathbf{x} \in C_i$ iff $f(\mathbf{x})$ returns the same label for all $\mathbf{x} \in C_i$.*

**Proposition 1** (Choice of Normalizer)**.** *Let $\mathcal{N}$ be a function that maps an input $\mathbf{x} \in C_i$ to any deterministic element in $C_i$. Then $f(\mathcal{N}(\cdot))$ is robust to $\mathcal{R}$-relational adversaries.*[2]

## 4.3 ROBUSTNESS-ACCURACY TRADE-OFF

***Optimal Accuracy under N&P.*** Let $\mu(\mathbf{x})$ denote the probability mass of $\mathbf{x}$. The label of an input $\mathbf{x}$ may also be probabilistic in nature, therefore we use $\eta(\mathbf{x}, l) = \Pr(y = l \mid \mathbf{x})$ to denote the probability that $\mathbf{x}$ has label $l$. [3] Then the optimal robust accuracy using *N&P*, denoted by $Acc_{\mathcal{R}}^*$, is $\sum_{C_i} \max_{l \in \mathcal{Y}} \sum_{\mathbf{x} \in C_i} \mu(\mathbf{x})\eta(\mathbf{x}, l)$, which happens when $f(\mathcal{N}(\mathbf{x})) = \arg\max_{l \in \mathcal{Y}} \sum_{\mathbf{x} \in C_i} \mu(\mathbf{x})\eta(\mathbf{x}, l)$ for $\mathbf{x} \in C_i$. Intuitively, $f$ shall assign the most likely label of random samples in $C_i$ to all $\mathbf{x} \in C_i$.

***Price of Robustness.*** In *N&P*, the optimal robust accuracy depends on $\mathcal{R}$. We then observe the following fundamental robustness-accuracy trade-off: as the relation becomes more complicated, we may lose accuracy for enforcing invariant model predictions, and such loss is the price of robustness.

**Observation 2** (Robustness-accuracy trade-off)**.** *Let $\mathcal{R}'$ and $\mathcal{R}$ be two relations s.t. $\mathcal{R}' = \mathcal{R} \bigcup \{(\mathbf{x}, \mathbf{z})\}$, i.e. $\mathcal{R}'$ allows an extra transformation from $\mathbf{x}$ to $\mathbf{z}$ than $\mathcal{R}$. Let $C_{\mathbf{x}, \mathcal{R}}$ denote the WCC in $G_{\mathcal{R}}$ that contains $\mathbf{x}$, and $l_C$ be the most likely label of inputs in a WCC $C$. Then $Acc_{\mathcal{R}'}^* - Acc_{\mathcal{R}}^* \leq 0$ for all $\mathcal{R}, \mathcal{R}'$ pairs, and the equality only holds when $l_{C_{\mathbf{x}, \mathcal{R}}} = l_{C_{\mathbf{z}, \mathcal{R}}}$.*

The intuition is that the extra edge on the relation graph may join two connected components which are otherwise separate. As a result, a model under *N&P* will predict the same label for the two components, thus the accuracy on one component will drop if two components have different labels.

We further characterize three different levels of trade-offs (Figure 1). First, if two inputs $\mathbf{x}, \mathbf{z}$ have the same most likely label on $\mathcal{D}$, then the optimal accuracy under *N&P* is the same as before normalization, in other words, robustness is obtained *for free*. Second, if both $(\mathbf{x}, \mathbf{z})$ and $(\mathbf{z}, \mathbf{x})$ are in $\mathcal{R}$ but $\mathbf{x}, \mathbf{z}$ have different most likely labels, then the model with the highest natural accuracy, which predicts the most likely label of $\mathbf{x}$ and $\mathbf{z}$ respectively, do not have any robustness. In contrast, *N&P* achieves the optimal robust accuracy by predicting a *single* label — the most likely label of samples in $\{\mathbf{x}, \mathbf{z}\}$ — for both $\mathbf{x}$ and $\mathbf{z}$. Third, if $\mathbf{x}$ can only be one-way transformed to two inputs $\mathbf{z}_1, \mathbf{z}_2$ with different

---

[2]Appendix C.1 shows a decidable algorithm of realizing such an $\mathcal{N}$ given $G_{\mathcal{R}}$.

[3]For example, a ransomware and a zip tool may have the same static feature vector $\mathbf{x}$. The label of a randomly drawn $\mathbf{x}$ is probabilistic, and the probability depends on the frequency that each software appears.

most likely labels, then *N&P* may have suboptimal robust accuracy. An absolutely robust classifier need to predict the same label for $\mathbf{x}, \mathbf{z}_1$ and $\mathbf{z}_2$, while the classifier with the highest robust accuracy should predict the mostly likely labels for $\mathbf{z}_1$ and $\mathbf{z}_2$ if $\mathbf{z}_1, \mathbf{z}_2$ appear more frequently than $\mathbf{x}$.

## 5   COMPARING AND UNIFYING *N&P* WITH ADVERSARIAL TRAINING

*N&P* differs from the adversarial training — the most widely acknowledged defense mechanism against test-time adversary — in its objective and procedure. While each approach has its own limitation against relational adversaries, we show that they can complement each other and be unified into one framework that enjoys the benefits from both worlds.

***Comparative Advantages.***   The performance of adversarial training depends on the quality of the adversarial examples. However, we show in Proposition 2 that the inner maximization problem is in general computationally infeasible for relational adversaries.

**Proposition 2** (Hardness of Inner Maximization). *The inner optimization problem of adversarial training is PSPACE-hard for relational adversaries.*

Intuitively, the search space of a relational adversary can grow combinatorially with the number of transformations, and the proposition follows the classical results of reachability analysis in model checking (Kozen, 1977). The *N&P* framework, in contrast, solves a typical minimization problem, and thus reduces the computation complexity if an efficient normalizer exists. Meanwhile, we show in Appendix A.4 that robust accuracy can be achieved with a simpler model class on normalized inputs than on original inputs; reduced model complexity may also improve the sample efficiency of the underlying learning algorithm. On the other hand, *N&P* may incur excessive loss in accuracy to enforce robustness, for example, the last scenario in Figure 1, in which case, adversarial training will be a better choice for overall utility.

***A Unified Framework.***   Motivated by the above observations, we propose a unified framework: for a relation $\mathcal{R}$, we strategically select a subset $\mathcal{R}' \subset \mathcal{R}$ to normalize inputs, and adversarially train on the normalized inputs. Let $\mathcal{N}_{R'}$ denote the normalizer for $\mathcal{R}'$. Formally, the learner solves

$$\min_{f \in \mathcal{H}} \max_{A(\cdot)} \sum_{(\mathbf{x}, y) \in D} \ell \left( f, A \left( \mathcal{N}_{\mathcal{R}'}(\mathbf{x}) \right), y \right), \tag{2}$$

during training to obtain a minimizer $f^*$, and predicts $f^*(\mathcal{N}_{\mathcal{R}'}(\mathbf{x}))$ at test-time. The classifier $f^*$ will be robust to $\mathcal{R}'$-relational adversary, and have potentially higher robust accuracy than using *N&P* alone. In particular, if $\mathcal{R}'$ is *reversible* by Definition 3, then our unified framework preserves the optimal robust accuracy as shown in Theorem 1.

**Definition 3.** *A relation $\mathcal{R}'$ is reversible iff $\mathbf{x} \to_{\mathcal{R}'*} \mathbf{z}$ implies $\mathbf{z} \to_{\mathcal{R}'*} \mathbf{x}$ and vice versa.*

**Theorem 1** (Preservation of robust accuracy). *Let $f^*$ be the classifier that minimizes the objective of our unified framework over data distribution $\mathcal{D}$, and let $f^*_{adv}$ minimize the objective of adversarial training over $\mathcal{D}$. Then, in principle, $f^*(\mathcal{N}_{\mathcal{R}'}(\cdot))$ and $f^*_{adv}$ have the same optimal robust accuracy if $\mathcal{R}'$ is reversible.*

The proof can be found in Appendix A.5. In essence, Theorem 1 is a generalization of the second scenario in Figure 1, in particular, we extend the same principle applied to $(\mathbf{x}, \mathbf{z})$ to all possible pairs of inputs in the relational graph induced by $\mathcal{R}'$. Note that reversible relation is also common: if $\mathbf{z}$ is $\mathbf{x}$'s adversarial example, then $\mathbf{x}$ is also likely to be an adversarial choice of $\mathbf{z}$. Observation 2 and Theorem 1 provide a general guideline for selecting $\mathcal{R}'$: choose the reversible subset of $\mathcal{R}$ first, and then consider transformations that cause little drop in optimal robust accuracy.

Regarding the efficiency of normalization, we show in Appendix A.6 that the strongest adversarial example satisfies the requirment of Proposition 1, and thus can be used as the normal form. Therefore, in theory, *N&P* is at least as efficient as the optimal adversarial training. In practice, the normalizer we use in our empirical evaluation are all more efficient than adversarial training.

## 6   EXPERIMENT

We now evaluate the effectiveness of our unified framework against relational attacks. In particular, we seek answers to the following questions:

Table 2: Malware Detection: False Negative Rate (FNR) and False Positive Rate (FPR) on *Sleipnir*.

| | **Unified (Ours)** | | Adv-Trained | | Al-Dujaili et al. (2018) | | Natural | |
|---|---|---|---|---|---|---|---|---|
| | FNR(%) | FPR(%) | FNR(%) | FPR(%) | FNR(%) | FPR(%) | FNR(%) | FPR(%) |
| Natural | 5.0±0.4 | 11.9±1.2 | 5.8±0.9 | 12.1±1.2 | 6.4±0.5 | 10.7±0.3 | 6.2±0.6 | 10.0±0.6 |
| Adversarial | 5.5±0.5 | 11.9±1.2 | 27.9±8.2 | 12.1±1.2 | 89.9±7.8 | 10.7±0.3 | 100±0.0 | 10.0±0.6 |

1. Do relational attacks pose real threats to existing ML models?

2. How effective is our unified framework in enhancing robustness, and do the results corroborate with the theory?

We investigate these aspects over two real world tasks — malware detection and image classification. For each task, we identify relations that do not alter the essential semantics of the inputs. Our result shows that the models obtained from our unified framework has the highest robust accuracy compared to adversarially trained models and unprotected models.

## 6.1 MALWARE DETECTION

We evaluate a malware detection task on *Sleipnir*, a data set containing Windows binary API usage features of 34,995 malware and 19,696 benign software, extracted from their Portable Executable (PE) files using LIEF (Thomas, 2017). The detection is exclusively based on the API usage of a malware. There are 22,761 unique API calls in the data set, so each PE file is represented by a binary indicator vector $\mathbf{x} \in \{0, 1\}^m$, where $m = 22,761$. Note that this is the same encoding scheme adopted by Al-Dujaili et al. (2018). We sample 19,000 benign PEs and 19,000 malicious PEs to construct the training (60%), validation (20%), and test (20%) sets.

Existing $\ell_p$ norm based attacks are not applicable for relational adversaries. Meanwhile, exhaustive search over adversarial choices may be computationally prohibitive. Therefore, we propose two heuristic attack algorithms – GREEDYBYGROUP and GREEDYBYGRAD – to validate models' robust accuracy. Both algorithms are greedy and iterative in nature. Detailed algorithm descriptions are in Appendix C.2.

**GREEDYBYGROUP** takes a test input vector $\mathbf{x}$ and a maximum number of iterations $K$. In each iteration, it partitions $\mathcal{R}$ into subsets of relations $\mathcal{R}_1, \cdots, \mathcal{R}_m$, and finds the instance within the transitive closure of each $\mathcal{R}_i$ that maximizes the loss. These instances from all $\mathcal{R}_i$s are combined to create the new version of $\mathbf{x}^{adv}$. Notice the attack reduces to exact search if $\mathcal{R}$ is not partitioned.

**GREEDYBYGRAD** takes a test input vector $\mathbf{x}$, a maximum number $m$ of transformation to apply in each iteration, and a maximum number of iteration $K$. In each iteration, it makes a first-order approximation of the change in test loss caused by each transformation, and then applies the transformations with top $m$ approximated increases in test loss to create the new version of $\mathbf{x}^{adv}$.

***Relation and Attacks.*** The goal of an adversary is to evade a malware detector. A common strategy that (Al-Dujaili et al., 2018) also adopts is adding redundant API calls. This strategy can be described by an additive relation: $(\mathbf{x}, \mathbf{z}) \in \mathcal{R}$ *iff* $\mathbf{z}$ is obtained by flipping some $\mathbf{x}$'s feature values from 0 to 1. We also consider a new attacking strategy, which substitutes API calls with functionally equivalent counterparts. This strategy can be described by an equivalence relation: $(\mathbf{x}, \mathbf{z}) \in \mathcal{R}$ *iff* $\mathbf{z}$ is obtained by changing some of $\mathbf{x}$'s feature values from 1 to 0 in conjunction with some of $\mathbf{x}$'s other feature values changed from 0 to 1. With expert knowledge, we extract nearly 2,000 equivalent API groups described in Appendix C.3. We use three attack algorithms — GREEDYBYGRAD, GREEDYBYGROUP and the `rfgsm_k` additive attack presented by Al-Dujaili et al. (2018) — and consider the attack to be successful if any algorithm fools the detector.

***Model and Baselines.*** We compare four ML detectors. The **Unified** detector is realized using our unified framework in Sec 5: we normalize over the equivalence relation based on the functionally equivalent API groups, and then adversarially trains over the additive relation. The **Adv-Trained** detector is adversarially trained with the best adversarial example generated using GREEDYBYGRAD and the `rfgsm_k` additive attack (Al-Dujaili et al., 2018) as GREEDYBYGROUP is too computationally expensive to be included in the training loop. We also include the model proposed by Al-Dujaili et al. (2018), which is adversarially trained against only the `rfgsm_k` additive attack, and a **Natural**

Table 3: Image Classification: Classification accuracy on CIFAR10, same relation in training and testing. The first column specifies the attack parameters used in test-time. The parameters are in the form of ($\ell_\infty$-norm, PGD step size, PGD steps, number of hue-shifts). The models are adversarially trained using $(4/255, 2/255, 3, 20)$.

| | Unified (Ours) | Adv-Trained (Combined) | Adv-Trained (PGD only) |
|---|---|---|---|
| Natural | 73.4±1.0 | 73.7±1.2 | 78.6±0.7 |
| $(4/255, 2/255, 3, 20)$ | 54.9±1.2 | 50.0±0.8 | 28.5±0.6 |
| $(4/255, 2/255, 3, 200)$ | 54.9±1.2 | 49.1±0.6 | 28.3±0.6 |
| $(4/255, 2/255, 15, 20)$ | 54.3±1.1 | 48.6±0.9 | 27.9±0.6 |
| $(4/255, 2/255, 200, 20)$ | 54.2±1.2 | 48.2±0.9 | 27.8±0.6 |
| $(6/255, 2/255, 15, 20)$ | 42.4±1.4 | 34.5±1.1 | 17.4±0.8 |
| $(6/255, 2/255, 200, 200)$ | 42.3±1.5 | 33.8±1.0 | 17.2±0.8 |

Table 4: Image Classification: Classification accuracy on CIFAR10, relation in training is a subset of relation in testing. The attacker uses a 15-step PGD attack with $\ell_\infty$-norm $4/255$ and step size $2/255$, and randomly samples 500 combinations of hue, brightness and constrast adjustment factors.

| | Unified (Ours) | Adv-Trained (Combined) | Adv-Trained (PGD only) |
|---|---|---|---|
| Natural | 73.4±1.0 | 73.7±1.2 | 78.6±0.7 |
| Adversarial | 47.0±1.1 | 41.7±1.1 | 19.7±0.8 |

model with no defense. We use the same network architecture as Al-Dujaili et al. (2018), a fully-connected neural net with three hidden layers, each with 300 ReLU nodes, to set up a fair comparison. We train each baseline to minimize the negative log-likelihood loss for 20 epochs, and pick the model with the lowest validation loss. We run five different data splits.

***Results.*** As Table 2 shows, relational attacks are overwhelmingly effective to detectors that are oblivious to potential transformations. Adversarial examples almost always (>99% FNR) evade the naturally trained model, and also evade the detector in Al-Dujaili et al. (2018) most of the time (>89% FNR) as it does not consider API substitution. On the defense end, **Unified** achieves the highest robust accuracy: the evasion rate (FNR) only increases by $0.5\%$ on average. **Adv-Trained** comes second but the evasion rate is still $22.1\%$ higher. The evasion is mostly caused by GREEDYBYGROUP, the attack that is too computationally expensive to be included in the training loop. This result corroborates with the theoretical advantage of *N&P*: its robustness guarantee is independent of training algorithms. Last, all detectors using robust learning techniques have higher FPR compared to **Natural**, which is expected because of the inevitable robustness-accuracy trade-off. However, the difference is much smaller compared to the cost due to attacks, and thus the trade-off is worthwhile.

## 6.2 IMAGE CLASSIFICATION

We evaluate the effectiveness of our unified framework on CIFAR10 containing 50,000 training and 10,000 test images of size 32x32 pixels. We randomly sample 5,000 images for validation and train on the remaining 45,000 images.

***Relation and Attacks.*** We consider a relation induced by hue shifting specified in Example 1. Due to the shape bias property (Landau et al., 1988), humans can still correctly classify most images after the adjustment of color hue. Therefore, we consider this relation to be semantics-preserving. The attacker uses a combination of $\ell_\infty$ and relational attacks: it first shifts the color hue of the image, and then generates $\ell_\infty$ adversarial example using PGD attack. For each image, the attacker tries different hue adjustments, which evenly split the hue space. In addition, we consider an attacker that can also adjust the brightness and contrast of the image by a factor in $[0.8, 1.2]$. It tries 500 random combination of hue, brightness and contrast adjustments followed by PGD attack.

***Model and Baselines.*** The **Unified** classifier is obtained with our unified framework in Sec 5: we adjust hue of the input such that the pixel at the top-left corner has hue value 1, and then adversarially

train against the PGD attack. [4] We also consider two adversarial training baselines: the first uses the combined attack (PGD and hue adjustment) in training, while the second only uses the PGD attack. We train a ResNet32 network for 100 epochs in all configurations, and pick the model with the lowest validation loss. We also run five different data splits.

***Results.*** Table 3 shows the results against attackers using hue-shift and $\ell_\infty$ perturbation. Although adversarial training against only the PGD attack has higher clean input accuracy, the combined attack heavily reduces its test accuracy, indicating again the effectiveness of simple relational attack to unprotected models. **Unified** achieves the highest robust accuracy against the combined attack – $\geq 4.8\%$ higher compared to adversarial training with the combined attack over all attack parameters. This result shows the advantage of normalization over reversible relations, as projected by our analysis in Sec 5. In addition, Table 4 shows the results against an attacker using more transformations than the ones normalized in training. Our unified approach still achieves the highest accuracy with a substantial margin over the baselines. Although the attacker may use more transformations, normalization can still reduce the search space of adversarial examples and increase robustness.

# 7 CONCLUSION AND FUTURE WORK

In this work, we set the first step towards robust learning against relational adversaries: we theoretically characterize the conditions for robustness and the sources of robustness-accuracy trade-off, and propose a provably robust learning framework. Our empirical evaluation shows that a combination of input normalization and adversarial training can significantly enhance model robustness. For future work, we see automatic detection of semantics-preserving transformation as a promising addition to our current expert knowledge approach, and plan to extend the normalization approach to deal with other kinds of attacks beyond relational adversaries.

---

[4]Given an input image in RGB format, we first convert the image to HSV format, and then add a scalar to the hue of all pixels. The scalar is determined by 1 - (the hue of the pixel on the first row and first column). The hue values are then projected back to the [0,1] interval by taking the remainder over 1. Finally, we convert the image back to RGB for classification.

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
