# OpenReview forum: "Robustness against Relational Adversary"
_ICLR.cc/2021/Conference — Reject_

### Official Review · AnonReviewer1 · 2020-10-28
**Interesting formulation to defend against adversarial attacks**

**Rating:** 6
**Confidence:** 2

**Review:**

The paper proposes to define robustness against adversarial attacks using a binary relation over inputs. The binary relation R states which input examples should be classified the same way and each connected component defined by R needs to be assigned the same label leading to robust classification. The idea of using a relation to capture what information in the input can be exploited for classification seems simple but was insightful to this reviewer. The authors propose to combine this approach of normalizing the input with existing techniques to handle adversarial classification to propose a new way to derive a robust classifier. Experiments on malware data and image data show that it leads to the best trade-off between classification accuracy and robustness.

In the experiments, the normalizers used to train the classifier are the same ones used to manipulate the test input. Is this realistic? Shouldn't it be the case that different techniques be used to manipulate the inputs for testing? I mean in section 6.1 Unified detector is trained using functionally equivalent API calls and the same is also used to launch an attack. Similarly, in section 6.2 hue shifting is used to manipulate test inputs while Unified classifier normalizes by setting the top left corner pixel to unit hue. Assuming such perfect knowledge of the adversaries techniques and using it to train a robust classifier seems unrealistic. How do the proposed techniques apply in more realistic situations where we would not know the adversary's techniques? How do we choose a normalizer in such situations? IMHO these questions need to be answered in the paper to ensure applicability in the real world, otherwise it seems this formulation of robustness to adversarial attacks is simply of academic interest.

Writing wise, the paper is an easy read. Proofs and other details are delegated to the appendix while the matter is made accessible by insightful, approachable, hand-holding discussions in the main body of the paper expressing the core ideas easily.

---

> ### Author Response · Authors · 2020-11-16
> **Responses to AnonReviewer1**
>
> We thank R1 for the valuable input. In fact, R1 raises a very insightful question: can the attacker possibly use transformations beyond the defender’s knowledge, and if yes, does this scenario hinder the use of N&P in practice? We acknowledge that the attacker indeed can use more transformations than considered in N&P in training time.
> However, the defender always needs to assume a relation in order to reason robustness against relational adversaries. Moreover, N&P can still contribute significantly to overall model robustness against real-world attacks as it can readily integrate with other defense mechanisms that deal with potential unknown transformations or different types of perturbations.
>
> First, we understand R1’s concern: in general, a strong defense mechanism shall make as few assumptions of the attacking strategy as possible to withstand the arms race. A famous warning example is that many defenses against $\ell_p$ perturbation are designed for gradient-based attacks, and are subsequently broken by CW attack.  However, assuming a set of admissible transformations is fundamentally different from assuming specific attack algorithms: the relation determines the *feasible set* of adversarial choices rather than the actual attack methods. Such feasible set corresponds to the $\ell_p$ ball for $\ell_p$-norm based attacks, and is absolutely needed for defining robustness.
>
> Second, our interests in relational adversary stem from resolving real-world challenges. As we show in our experiment, even simple relations such as API substitution and hue-shift can already substantially reduce the model performance. Meanwhile, little is understood on systematically dealing with such relational attacks even when the defender knows the relation. There is a gap between using black-box ML models v.s. incorporating robustness properties derived from expert knowledge. Our work proposes a framework to bridge the gap, and we see exploring domain knowledge and designing fast normalization procedures an exciting future direction.
>
> Last, we believe the N&P can significantly contribute to model robustness against complex real-world attacks. We are clearly aware that N&P is not the perfect solution to every attack. In fact, the trade-off analysis in Sec 4.3 and the comparison with adversarial training in Sec 5 are exactly addressing the question of when to use N&P. The key message is that N&P can provide robustness guarantee to some attack behaviors, and also reduces the search space complexity of adversarial training. It can always complement rather than replace existing techniques. Our empirical results show that normalizing over simple relations can already improve the performance over existing baseline. In conclusion, we believe adversarial semantic-preserving transformation as a crucial challenge to ML, and we see N&P play an important role in the defense pipeline.

---

> > ### Comment · AnonReviewer1 · 2020-11-18
> > **Are experiments illustrating sensitivity towards the normalization operator feasible?**
> >
> > I appreciate your response. I definitely agree with your second point about little being understood on systematically treating relational attacks and that N&P extends the state-of-the-art in this respect.
> >
> > I am wondering whether it is possible to design an experiment that can show what happens when a normalization operator slightly different from the one used by the attacker, is used by the defender. For instance, instead of normalizing the top left corner to be of unit hue, how about we normalize something else (E.g., luminosity) ? It would be nice to know if the robust classifier thus learned can still provide some level of accuracy. And even if its not, it would be nice to know how bad it can get just so the reader knows how sensitive N&P is to the normalization operator.

---

> > > ### Author Response · Authors · 2020-11-19
> > > **Additional experiment and discussion over the discrepancy b/w training and test time relation**
> > >
> > > Thanks for the discussion! We see your concern over the discrepancy between the relation used by the learner and the attacker, and we totally agree that such discrepancy can happen in real-world tasks. While a rigorous definition or measure of "sensitivity" is hard at the moment, we design a new experiment in which the learner only knows a subset of the attacker's transformations during training. The result shows that our unified framework with such partial information still improves the model robustness compared to adversarial training alone.
> > >
> > > First, the notion of sensitivity often refers to how much the outcome may change w.r.t. changes in an underlying factor. However, it's hard to measure sensitivity of the performance of N&P w.r.t. change in test-time relation because the change in relation is hard to quantify. Intuitively, there's not a good numerical way to measure, e.g., the difference between luminosity adjustment v.s. hue shift, and therefore it's unclear what robust accuracy change to expect.
> > >
> > > Despite the difficulty, we design an experiment to investigate a common scenario: the relation in training is a subset of the relation in test. For the CIFAR10 task, the learner still normalizes over the hue space. The attacker, however, can scale the brightness and saturation by a factor in [0.8, 1.2] on top of the $\ell_{\infty}$ attack and hue-shift. The results are shown in the table below.
> > >
> > > | | |
> > > |:---------------:|:---------------:|
> > > |Unified (Ours)|$\hspace{10pt} 47.0\pm 1.1$|
> > > |Adv. Trained over PGD+hue shift|$\hspace{10pt} 41.7\pm 1.1$|
> > > |Adv. Trained over PGD only|$\hspace{10pt} 19.7\pm 0.8$|
> > >
> > > The robust accuracy of all three methods drops compared to the value in Sec 6 of our paper. This is expected as the attacker can search adversarial examples from a larger feasible set. However, our unified framework -- which only normalizes over the hue space -- still has better robust accuracy than the two adversarial training baselines.  Using N&P in the defense pipeline can reduce the search space of adversarial examples. Such advantage still holds even when the normalization is only performed on a subset of the relation in test-time.
> > >
> > > In addition, we note that N&P may also choose to only normalize over a subset of the relation for better robustness-accuracy trade-off in practice. Our theoretical analysis has a thorough discussion on the choice of relation.
> > >
> > > We hope this new experiment can answer your question, and we are happy for more follow-up discussion. We thank R1 again for the feedback that helps improve our paper!

---

> > > > ### Comment · AnonReviewer1 · 2020-11-23
> > > > **The new experiment is a valuable addition**
> > > >
> > > > I thank the authors for designing and running the experiment. It doesn't answer all my questions, but given the difficulty of the ask I think the authors are to be commended.
> > > >
> > > > PS: I have bumped up my rating.

---

### Official Review · AnonReviewer3 · 2020-10-28
**Official Blind Review #3**

**Rating:** 7
**Confidence:** 4

**Review:**

This work investigates relational adversary which creates adversarial examples by transforming the original inputs via a logical relation. The authors analyze the conditions for robustness to relational adversary and propose a learning framework which learns and predicts over normalized training and test inputs. Since their framework may incur excessive loss in accuracy, the authors then propose a unified framework by combining input normalization with adversarial training and claim that their unified framework can preserve the optimal robust accuracy. Empirical results show that their unified framework can enhance model robustness on two real-world tasks.

Overall, I vote for accepting the paper. I like the idea of relational adversary and the technique of selecting a reversible relation for input normalization in their unified framework. My main concern is about the clarity of the paper, especially its proofs. (see cons below).

Pros:
1. The paper considers relational adversaries which create adversarial examples in a semantics-preserving manner by a logical relation. I think this problem is practical and important to the adversarial learning community.
2. The proposed normalization procedure which guarantees robustness against relational adversaries is promising and might be applied to other kinds of adversaries. The robustness-accuracy trade-off analysis is reasonable, and the design for combining $N$\&$P$ with adversarial training is also interesting.
3. This paper provide experiments on two real world tasks to corroborate with their theory. The performance of their framework is good.

Cons:
1. I have some questions about the proof of Theorem 1 in A.5.
- In the proof of Claim 3, the authors denote $z$ such that $N_{R’}(x)\to_ {R\\R’}^* z$. Then they show that $N_{R’}(x)\to_{R\\R’}^* z$ implies $x\to_{R}^* z$. However from the assumption that f is not robustly accurate over $(x, y)$ under $R$, it can be directly derived that there exists $z$ such that $x\to_{R}^* z$ and $f(z)\neq y$. Why the authors use the extra step?
- In the last three line of A.5, the authors state that “f* is at least as robustly accurate as f”. This statement is a little confusing. If I understand correctly, the authors want to say that f* has at least the same robust accuracy as f. Is that right? Moreover, why the statement “f* has at least the same robust accuracy as f” is true? Since it is actually not so obvious to the readers, can the authors provide rigorous proofs?
- The notation $\to_{R\\R’}^*$ is misleading. I guess what the authors want to say here is that $x\to_{R\\R’}^* z$ iff $x\to_{R}^* z$ and $x\not \to_{R’}^* z$. Moreover, the notation $A(\cdot)$ used in the first formula of Theorem 3 is different from that of the second formula. It would be better if the authors can point it out.

2. There are some questionable inaccuracies in the proof of Claim 2 in A.4.
-  what does the symbol $\vee_{i\in U}$ in Definition 4 mean? Taking a maximum over $i\in U$?
- In Claim 2, the authors state that $x_1$ and $x_1’$ satisfy the equivalence relation in Definiton 4 (what is U here?). However, it seems that $x_1, x_1’, x_2, x_3, x_4$ are components of the d-dimensional vector $x$, that is, all $x_i$s are scalar. Moreover, the sentence “iff any of the following is true:” is confusing. This sentence seems not to fit in the context well.
- Can the authors explain more details on why the classifier must satisfy (5), (6), (7) and (8)?

3. In Proposition 4 in A.6, the symbol $y$ in $l(f, z, y)$ is misleading. The readers might think that $y$ depends on $x$. In fact, here $y$ should be fixed for all $x\in C_i$, isn't it?


Some typos:

In the last four line of second paragraph of Section 2, “against to”-> “against”.

In the last line of Page 12, “Observation 2”-> “Proposition 2”.

In Theorem 2, $\\{0, 1\\}^n$-> $\\{0, 1\\}^d$.

In the second line of A.7 in Page 15, “bizzare”->”bizarre”.

---

> ### Author Response · Authors · 2020-11-14
> **Responses to AnonReviewer3**
>
> We are glad that the reviewer is interested in the study of robustness against relational adversarial, and we truly appreciate the reviewer's careful examination over the theories. We will incorporate the reviewer's feedback to improve the presentation of our proofs. Here, we focus on clarifying the confusion over the notation and proof steps.
>
> ***Proof of Theorem 1***
>
> Recall that both $\mathcal{R}$ and $\mathcal{R}’$ are sets of admissible transformations. The notation $\mathcal{R}\backslash\mathcal{R}’$ denotes the set of transformation in $\mathcal{R}$ but not in $\mathcal{R}’$, i.e. the set difference. Notice that $\mathcal{R}\backslash\mathcal{R}’$ is a relation too, and $\mathbf{x}\to_{\mathcal{R}\backslash\mathcal{R}’}^* \mathbf{z}$ means $\mathbf{x}$ can be transformed to $\mathbf{z}$ using rules in $\mathcal{R}\backslash\mathcal{R}’$.
>
> The proof can indeed be shortened as the reviewer suggests. By assuming $f$ is not robustly accurate over some $(\mathbf{x}, y)$, we have established a $\mathbf{z}$ such that $\mathbf{x}\to_{R}^* \mathbf{z}$ and $f(\mathbf{z})\neq y$.
>
> Regarding the robust accuracy of $f^*$ v.s. $f$, the reviewer is correct; we meant $f^*(\mathcal{N_{\mathcal{R}'}}(\cdot))$ has at least the same robust accuracy as $f$. The reason is the following. $f^*(\mathcal{N_{\mathcal{R}'}}(\cdot))$ is the classifier with the highest robust accuracy in our unified framework. By definition, it has at least the same robust accuracy $f(\mathcal{N_{\mathcal{R}'}}(\cdot))$. Meanwhile, by the construction of $f$, we have $f(\mathcal{N_{\mathcal{R}'}}(\mathbf{x})) = f(\mathbf{x})$ for all $\mathbf{x}$. Therefore, $f$ has the same robust accuracy as $f(\mathcal{N_{\mathcal{R}'}}(\cdot))$, and the statement holds.
>
> To complete the proof, we have shown in Claim 3 that $f$ has at least the same robust accuracy as $f_{adv}^*$, so $f^*(\mathcal{N_{\mathcal{R}'}}(\cdot))$ has at least the same robust accuracy as $f_{adv}^*$. Meanwhile, $f_{adv}^*$ has the highest robust accuracy by definition. Therefore, $f_{adv}^*$ and $f^*(\mathcal{N_{\mathcal{R}'}}(\cdot))$ has the same robust accuracy.
>
>
> ***Proof of Claim 2 in A.4***
>
> First, each feature $\mathbf{x}_i$, a scalar, is a component of the input vector $\mathbf{x}$. In terms of the equivalence condition, $\mathbf{x}_i$ is also a binary variable that takes value $0$ or $1$. We apologize for this confusion, and will carefully clarify in our revision.
>
> The notation $\bigvee_{i\in U} \mathbf{x}_i$ means taking a *logic or* operation over $\mathbf{x}_i$ for all $i$ in the set $U$. For example, if $U=[1,5]$, then the result will be $\mathbf{x}_1 \vee \mathbf{x}_5$. By $\mathbf{x}_1$ and $\mathbf{x}’_1$ satisfying the equivalence relation, we mean $U = [\mathbf{x}_1, \mathbf{x}’_1]$. Note that in Definition 4, $U$ refers to the set of indices of the equivalent components. We overload the notation here to represent the set of the actual equivalent components. We will make this point clearer in our revision.
>
>
> In Claim 2, the ground truth label $y=1$ if and only if any of the three listed clauses is true. For example, if $\mathbf{x}_2 = 1$ and $\mathbf{x}_3 = 1$, then $y=1$ because Clause 1 is true. Similarly, if $\mathbf{x}_1 = 1$ and $\mathbf{x}_2 = 1$, then $y=1$ because Clause 2 is true. On the other hand, if $\mathbf{x}_2 = 1$, $\mathbf{x}_4 = 1$ and the other features are $0$, then $y=0$ because none of the clauses are true.
>
> Equation 5-8 then naturally follows from the labeling of $y$. Suppose $\mathbf{w}$ is a robust and accurate classifier, and we take $0$ as the threshold, i.e. prediction score $>0$ means $y=1$ and $<0$ means $y=0$. Equation 5 means the linear classifier must have a score $<0$ when only $\mathbf{x}_2$ and $\mathbf{x}_4$ have value $1$. Similarly, Equation 6 means the linear classifier must have a score $>1$ when only $\mathbf{x}_1$ and $\mathbf{x}_2$ have value $1$. No $\mathbf{w}$ can satisfy all equations at the same, and therefore a robust and accurate linear classifier does not exist.
>
>
> ***Proposition 4 in A.6***
>
> In Proposition 4, $\mathbf{x}$ is an input instance, $y$ is its true label, and $\mathbf{z}^*$ is the adversarial example that maximizes the adversarial loss $\ell(f, \mathbf{z}, y)$. The tuple $(\mathbf{x}, y)$ will be a sample point drawn from the data distribution.
>
> We would like to thank the reviewer again for the thorough examination of the proof details, and we are happy to answer any follow-up questions.

---

### Official Review · AnonReviewer5 · 2020-11-07
**New framework and angle to study adversarial robustness; more empirical evidence needed**

**Rating:** 6
**Confidence:** 5

**Review:**

This paper proposed a new framework, relational adversary, to study adversarial robustness under multiple (sequential) data transformations. The authors also made robustness guarantees based on the Normalizle-and-Predict scenario. Improved robustness is numerically presented on two tasks: malware detection (under add/substitute attack) and image classification (under joint hue and Linfinity perturbation).

Strengths:
1. The relation adversary perspective is (relatively) novel in studying adversarial robustness
2. The same framework can apply to very different modalities, such as malware and image

Weaknesses:
1. The numerical results (especially on image classification) should be expanded to better support the robustness claim
2. The normalize-and-predict part sometimes lacks clarity, and it's unclear how to select the normalization function

Detailed Comments:
Overall, the relational adversary framework, in my opinion, adds some novel insights for studying adversarial robustness. The malware detector experiment setup and performance better demonstrates the utility compared to the image classification results.

I have some concerns about the current version, which are detailed below.

1. For image classification task against combined Linfinity pixel perturbation and hue shift attack, it appears that the authors use the same attack hyperparameters in training and testing. Will the advantage in the proposed method over adversarial training still exist for different train/test hyperparameter settings? Since there are only two attack types, I suggest the authors try different parameters on Linfinity threshold and hue changes when testing the robustness and report the robustness trend in the full spectrum.

2. If I understood correctly, this relation adversarial setting also applies to the multiple-perturbation setting, where each perturbation type can be viewed as one "relation". There are several existing works on multiple-type attacks, such as simultaneous/sequential perturbation with different Lp norms [R1, R2]. How will the proposed framework perform in multiple Lp-norm perturbation setting, and how does it compare to existing works?

[R1] Adversarial Training and Robustness for Multiple Perturbations, NeurIPS 2019
[R2] Adversarial Robustness Against the Union of Multiple Perturbation Models, ICML 2020

3.  It seems difficult or unclear to select a proper normalization function for a given task. For image classification, it was not clear what "normalization" means in this concept. It was clearly mentioned in the malware detection setting but not in image classification. Please clarity how a normalization function is determined and how easy the principle generalizes across different tasks.

---

> ### Author Response · Authors · 2020-11-16
> **Responses to AnnoReviewer5, Continued**
>
> ***Q: How will the proposed framework perform in the multiple $\ell_p$-norm perturbation setting, and how does it compare to existing works?***
>
> This is an excellent question. We are aware of the multiple perturbation literatures. We do not compare with them because our relational adversaries in general are orthogonal to the $\ell_p$-norm based attackers. Recall that our relational adversaries search adversarial examples in the transitive-closure of the relation $\mathcal{R}$. Composition of multiple transformations shall still preserve the semantics. In contrast, composing $\ell_p$-norm perturbation may not preserve semantics in *our* setting. For example, adding perturbation with $\ell_p$-norm $\epsilon$ twice may cause the adversarial example to be $2\epsilon$ away from the original and look substantially different. In fact, arbitrary composition of $\ell_p$-norm perturbation can transform an input $\mathbf{x}$ to any $\mathbf{z}$ in $\mathbb{R}^d$, and clearly, such transformations are no longer semantics preserving. Therefore, our notion of relational adversary is not immediately applicable to $\ell_p$-norm attacks.
>
> Despite the difference, one main message of our paper is that N&P can always complement rather than conflict with the existing adversarial training techniques. In fact, in the CIFAR10 experiment, we use normalization for hue-shift and adversarial training for the $\ell_p$-norm attack. The combined approach not only gives guaranteed robustness to hue-shift but also significantly reduces the search space of adversarial training. We believe that robust learning against semantic-preserving perturbation (e.g. relational adversaries) and syntactical-based perturbation (e.g. $\ell_p$-norm) are two parallel tracks that can work in tandem in practice to solve real-world challenges.
>
> ***Q: How a normalization function is determined, and how easy the principle generalizes across different tasks?***
>
> The normalization function for image classification against hue-shift is briefly mentioned in Model and Baselines in Sec 6.2. We apologize for the confusion, and will elaborate that in the paper. Given an input image in RGB format, we first convert the image to HSV format, and then add a scalar $\delta$ to the hue of all pixels. The scalar $\delta$ is determined by 1 - (the hue of the pixel on the first row and first column). The hue values are then projected back to the [0,1] interval by taking the remainder over 1. Finally, we convert the image back to RGB for classification.
>
> Here is a running example: suppose the input image is 3 pixels wide and 1 pixel tall, and the hue values are [0.5, 0.1, 0.9]. We will add 0.5 to the hue of all pixels and take the remainder over 1. The resulting image will have hue [1, 0.6, 0.4].
>
> We acknowledge that normalization can be hard for certain relations. In Appendix C.1, we provide a generic normalizing procedure given the relational graph. In practice, finding a normalizer requires domain knowledge. In our work, we can efficiently find the normalizer in a congruence group, e.g. source code using equivalent APIs, images that are hue-shifted from each other. Efficient normalizers also exist for data structures like executables [1], context-free grammars (i.e., Chomsky Normal Form), etc. Last, We show in Appendix A.6 that for reversible relation, the strongest adversarial example itself is a normal form, so normalization is no slower than adversarial training. On the brighter side, our main message is that using N&P whenever an efficient normalizer exists already outperforms adversarial training alone. We find exploring domain knowledge and designing fast normalization procedures an exciting future direction.
>
> [1] A Generic Approach to Automatic Deobfuscation of Executable Code

---

> > ### Comment · AnonReviewer5 · 2020-11-17
> > **Thank you for the clarification**
> >
> > I thank the reviewers for the clarification. My concerns in this part are fully resolved. I hope the reviewers can revise the submission accordingly. I will adjust my score once the attack results with more steps are reported.

---

> ### Author Response · Authors · 2020-11-16
> **Responses to AnnoReviewer5**
>
> Thank you for the valuable feedback. We are glad to see your interests in robustness against relational adversaries, and would like to clarify and resolve the main concerns in the review.
>
> ***Q: Will the advantage in the proposed method over adversarial training still exist for different train/test hyperparameter settings?***
>
>
> The key attack hyperparameters are: max $\ell_{\infty}$-norm constraint, step size, number of iterations for PGD attacks, and number of hue-shift attempts. We’ve run additional experiments with varying attack hyperparameters in test-time, and our results show that the advantage remains under increasing attack strength.
>
> First, we vary the PGD attack step size and number of iterations, while keeping the $\ell_{\infty}$ constraint and hue-shift attack constant. The result is presented in the following table: each column header represents a (step size, number of iterations) combination. In training, we use (2/255, 3). We see that for all combinations, our unified approach significantly outperforms adversarial training.
>
> |    	| $\hspace{10pt}$(2/255, 3)$\hspace{10pt}$ | $\hspace{10pt}$(2/255,7)$\hspace{10pt}$ | $\hspace{10pt}$(2/255,15)$\hspace{10pt}$ | $\hspace{10pt}$(1/255, 7)$\hspace{10pt}$ | $\hspace{10pt}$(1/255,15)$\hspace{10pt}$ |
> | :-----------: | :-----------: |:-----------:|:-----------:|:-----------:|:-----------:|
> |	Unified(Ours) 	| $54.9\pm1.2$  |$54.3\pm1.2$|$54.3\pm1.1$|$54.5\pm1.2$|$54.2\pm1.3$|
> |	Adv. Trained Only  | $50.0\pm0.8$ |$48.3\pm0.9$|$48.6\pm0.9$|$49.1\pm0.8$|$48.4\pm0.9$|
>
> Second, we increase the $\ell_{\infty}$ norm constraint from $4/255$ to $6/255$. The result is presented in the following table. The robust accuracy drops for both methods as the attacker is stronger than the one used in training. Nonetheless, our unified method still outperforms adversarial training by a substantial margin.
>
> |    	| $\hspace{10pt}$(2/255, 7), $\epsilon$=6/255 $\hspace{10pt}$|
> | :-----------: | :-----------: |
> |	Unified(Ours) 	| $42.8\pm1.3$  |
> |	Adv. Trained Only  | $35.4\pm1.0$ |
>
> Last, we increase the number of hue-shift the attacker can use from 20 times in training to 200 times. The result is shown in the table below. Both methods are fairly robust to the mere increase in hue-shift attempts. however, adversarial training still has a slight dip in performance, while our unified method is unaffected.
>
> |    	| $\hspace{20pt}$20 $\hspace{20pt}$| $\hspace{20pt}$200 $\hspace{20pt}$
> | :-----------: | :-----------: | :--------------:|
> |	Unified(Ours) 	| $54.9\pm1.2$  |$54.9\pm1.2$|
> |	Adv. Trained Only  | $50.0\pm0.8$ |$49.1\pm0.6$|
>
> Overall, our unified framework outperforms adversarial training over all attack parameter choices. The consistent advantage suggests that adversarial training has a difficulty in finding a good solution to the min-max optimization problem w.r.t. the relational adversaries. N&P is a valuable addition as it ensures robustness w.r.t. the relations being normalized and also reduces the search space for adversarial training.

---

> > ### Comment · AnonReviewer5 · 2020-11-17
> > **Please add attack results with more attack steps**
> >
> > I thank the authors for providing additional results. They are very insightful and I am very happy to see them being discussed and presented. To further validate the results, I would like to see more attack results with increased attack steps (say 100, 200, 500), to ensure the robustness gain is not coming from insufficient attack steps.

---

> > > ### Author Response · Authors · 2020-11-18
> > > **Result with More Attack Steps**
> > >
> > > We run PGD attacks with more steps. The results are shown in the table below.
> > >
> > > | |$\hspace{10pt}$ (2/255, 100) $\hspace{10pt}$| $\hspace{10pt}$ (2/255, 200) $\hspace{10pt}$|$\hspace{10pt}$ (1/255, 500) $\hspace{10pt}$|
> > > |:----------:|:----------:|:----------:|:----------:|
> > > |Unified (Ours)|$54.3\pm 1.2$|$54.2\pm 1.2$|$54.3\pm 1.2$|
> > > |Adv. Trained Only|$48.2\pm 0.9$|$48.2\pm 0.9$|$48.2\pm 0.9$|
> > >
> > > Overall, the difference is minor compared to using 15 steps and negligible for more than 100 steps. As a sanity check, we run pure PGD without hue-shift and the same trend holds. We thank the reviewer for the suggestion and we are happy to answer any follow-up question you may have.

---

> > > > ### Comment · AnonReviewer5 · 2020-11-18
> > > > **Thanks!**
> > > >
> > > > Thank you! I've increased my score accordingly.

---

### Official Review · AnonReviewer4 · 2020-11-07
**ICLR 2021 Conference Paper157 AnonReviewer4**

**Rating:** 4
**Confidence:** 5

**Review:**

## Summary

The authors investigate relational adversaries, adversarial attacks that
lie in a reflexive-transitive closure of logical relation. The authors
focus on such attacks as they are motivated by attacks that are not
necessarily bound to small lp-norms (typical of security application
domains). The authors analyze the condition of robustness required in
this context and propose normalize-and-predict, a learning framework
with provable robustness guarantees. The authors further compare their
approach against adversarial training, which helps them to propose a
unified theoretical framework. While relational adversary attacks seem
successful against adversarial training models, the authors' framework
enhances their robustness.

## Strengths

+  Adversarial ML attacks that are not necessarily bounded by small
lp-norms are interesting and still not widely studied
+  Interesting intuition around the concept of normalization

## Weaknesses

-  Effectiveness of normalization is bound to the set of semantics-preserving transformations available to attackers. How easy is it to find a normalizer? How easy is it for attackers to find another set of semantics-preserving transformations that would void the normalizer?

-  Likely experimental bias in the malware experiments?

## Comments

I found the paper well-written and motivated. I particularly liked the
fact the authors focus on adversarial attacks those that are not
necessarily bounded by small lp-norms (e.g., adversarial attacks "in the
problem space" [1]).

The authors explore an interesting problem that's been often neglected
but that plays a fundamental role in specific application domains (e.g.,
security). It would have been interesting to see how the authors' work
compared against a recently-published work [1] that reasons along the
same line. In particular, I wonder whether the theoretical framework in
[1], which reformulate adversarial ML attacks at test-time in the
problem space would overlap the authors' contributions or would
represent an orthogonal contribution.

The normalize-and-predict learning framework relies on the ability of
identifying semantics-preserving transformations, embedded in logical
relation. How hard is it to find such semantics-preserving
transformation? Are you assuming those are known/available to the
attacker? Would these transformations include side-effects depending on
the abstraction (program analysis) one relies on to extract features of
interest? Overall, I am quite intrigued by the benefits of the authors'
unified framework that normalizes over relations that preserve semantics
and adversarially train over the rest. However, from a practical
perspective, one would need to consider several other constraints when
reasoning on specific application domains (e.g., malware). Although I
value the authors' theoretical findings, I am unsure whether the
experiments support such claims thoroughly. I'd be more at ease if the
authors had reasoned on the robustness of the semantics-preserving
transformation they propose (besides the fact these really depend on the
underlying program analysis abstraction the classifier relies on), as
outlined further in [1].

Don't get me wrong: this is an interesting paper that explores the
effectiveness of the authors' defense. However, I believe it's worth
understanding whether the attack the authors propose is realistic or
there are any loose ends that are better formalized in [1]. If so, it
would be more appropriate to understand how the authors' defense is
robust against adversarial ML attacks at test-time in the problem space
(realizable attacks). More importantly, tho, I am concerned about the
practicality of the approach. It is clear that if an object x and its
corresponding adversarial object x_adv share the same normal form, the
prediction will be robust (as the authors rightly point out in Section
4.2). But how easy is it to find such a normalizer? This is strictly
dependent on the set of semantics-preserving transformation attackers
have at their disposal. In a way, this is exactly what [1] points out
and consider as robustness to pre-processing.

The authors may want to see [2] to make sure to remove any spatial bias
that may inflate the end results (in the security domain, it is common to expect
class imbalance as it is representative of real-world settings).

[1] https://s2lab.kcl.ac.uk/projects/intriguing/ (IEEE S&P 2020)

[2] https://www.usenix.org/system/files/sec19fall_pendlebury_prepub.pdf (USENIX Sec 2019)

## Edit after discussion

I thank the authors for the clarification. I still like the work and believe it is promising. However, I don't find too convincing the arguments around problem-space and malware experiments, I am afraid. Also, it seems there is still experimental bias in the evaluation (on malware), which hinders a bit the opportunity to assess the actual effectiveness of the authors' approach. I would really encourage the authors to reason about the points raised in the review in a more principled way.

---

> ### Author Response · Authors · 2020-11-13
> **Responses to AnonReviewer4**
>
> Thank you for the valuable input. We find [1] very interesting, especially the part of formalizing various real-world constraints to realizable attacks. Besides, we believe [1] will be very helpful to our learning framework in formulating the transformations. We are excited to see this concurrent work and will elaborate the shared spirit in the revised related work section.
>
> On the other hand, we feel our work and [1] are orthogonal and complementary despite the overlap in the adversarial settings: [1] focuses on realistic attacks on raw inputs, while our work focuses on developing defense mechanisms to adversarial transformations. The transformation can also be over both problem and feature space. Since there is no prior defense for general relational adversaries, we benchmark against the most widely used adversarial training. We show that a combined approach of our N&P and adversarial training has various advantages in theory and actually leads to higher robust accuracy than adversarial training in practice. One intriguing direction for future work will be applying our N&P defense strategy in the context of [1].
>
> ***Q: Who determines the admissible transformations, and how?***
>
> The transformations are derived from expert knowledge, e.g., in malware detection, we apply our knowledge to find APIs with the same functionality, and allow substitution among equivalent APIs. The rules also follow the semantic-preserving principle in [1]: they won’t change any malicious semantics of the software.
>
> We assume the attacker to have access to the transformations. However, we note that (1) robust learning even when transformations are available to the attacker is an unresolved challenge (c.f. first paragraph on Page 2), and (2) it’s unclear how predictions can be made robust to transformations beyond the learner’s knowledge base. The learner can always expand his/her knowledge base by adding more rules.
>
> ***Q: Will the defense mechanism be effective against problem-space attack?***
>
> According to [1], our malware detection experiment is performed on the API usage feature-space. The result against problem-space attack depends on both the quality of the feature extractor and the choice of rules. We characterize the possibilities into three cases. The saving grace is that the learner can always adjust the relation to avoid drop in performance.
>
> First, the extractor finds all API used, but there is no realizable malware instance corresponding to the feature-space adversarial example. Then the defense mechanism will remain effective against problem-space attack, as the attacker’s power is reduced because of the additional constraints.
>
> Second, the feature extractor finds all API used, but the attacker can transform on the problem-space so that the feature vector is outside the transitive-closure of R, then the defense can be less effective. However, the learner can always expand R. In our paper, the relation on the feature-space fully captures the transformations on problem-space.
>
> Third, the extractor fails to find all API, e.g. the attacker can mask API usage from popular extractors. The defense becomes less effective. However, it’s doubtful if any good API-based detector exists, as the feature vector no longer faithfully presents program semantics. In this case, the learner can choose a different representation and relation. Given an underlying distribution over the representation space, Sec 4.3 also states that the robust-accuracy trade-off is expected.
>
> ***Q: How easy is it to find a normalizer?***
>
> We acknowledge that normalization can be hard for certain relations. However, our main message is that using N&P whenever an efficient normalizer exists already outperforms adversarial training alone. In our work, we can efficiently find the normalizer in a congruence group, e.g. source code using equivalent APIs, images that are hue-shifted from each other. Efficient normalizers also exist for data structures like executables [2], context-free grammars (i.e., Chomsky Normal Form), etc. Last, We show in Appendix A.6 that for reversible relation, the strongest adversarial example itself is a normal form, so normalization is no slower than adversarial training. We find exploring domain knowledge and designing fast normalization procedures an exciting future direction.
>
> [1] https://s2lab.kcl.ac.uk/projects/intriguing/ (IEEE S&P 2020)\
> [2] A Generic Approach to Automatic Deobfuscation of Executable Code

---

### Decision · Program_Chairs · 2021-01-07
**Final Decision**

**Decision:**

Reject

**Comment:**

I thank the authors and reviewers for the lively discussions about the paper. All reviewers indicated that the work has merits and novelty however there were concerns about showing the benefits of the proposed method experimentally specially on malware applications. Given all, I think the paper needs a bit more work to be accepted.

- AC